# Understanding In-Context Learning from Repetitions

**Jianhao Yan**[1,2]  **Jin Xu**[4]  **Chiyu Song**[1,2]  **Chenming Wu**[5]  **Yafu Li**[1,2]  **Yue Zhang**[2,3,✉]

[1]Zhejiang University      [2]School of Engineering, Westlake University
[3] Institute of Advanced Technology, Westlake Institute for Advanced Study
[4] Tsinghua University      [5] Baidu Research
`elliottyan37@gmail.com`

## Abstract

This paper explores the elusive mechanism underpinning in-context learning in Large Language Models (LLMs). Our work provides a novel perspective by examining in-context learning via the lens of surface repetitions. We quantitatively investigate the role of surface features in text generation, and empirically establish the existence of *token co-occurrence reinforcement*, a principle that strengthens the relationship between two tokens based on their contextual co-occurrences. Furthermore, we find similar reinforcements lie behind the pretraining corpus, revealing the existence is due to LLMs' efforts to maximize likelihood. By investigating the dual impacts of these features, our research illuminates the internal workings of in-context learning and expounds on the reasons for its failures. This paper provides an essential contribution to the understanding of in-context learning and its potential limitations, providing a fresh perspective on this exciting capability.

## 1  Introduction

The impressive ability of Large Language Models (LLMs; Touvron et al. 2023a; Chowdhery et al. 2022; OpenAI 2023) to execute in-context learning (ICL) is a standout characteristic. This behavior mirrors human learning and reasoning from *analogy* (Winston, 1980), enabling LLMs to rapidly adapt to a range of downstream tasks. Without being explicitly pretrained to learn from demonstrations, LLMs can predict responses to unseen test queries from a few demonstrations and without any instruction given (Brown et al., 2020; Zhang et al., 2022; Chowdhery et al., 2022). An example of in-context learning can be found in Figure 1(a), where a pre-trained LLaMA model is given demonstrations for a binary classification task, and learns to make predictions correctly. Despite the success in applications, the working mechanism of in-context learning is still an open question.

Existing work has investigated input-label mapping (Min et al., 2022; Yoo et al., 2022; Wei et al., 2023) and demonstration construction (An et al., 2023; Lu et al., 2022; Liu et al., 2022) as underlying factors for ICL. However, little research has focused on the correlation between ICL and textual features. Intuitively, the behavior of ICL depends on the context and can be fragile to its variations. As Figure 1(b) shows, the same LLaMA model makes the incorrect prediction '*True*' given the input "*Circulation revenue has decreased by 5% in Finland.*", which is likely because of the repeated pattern "Answer:" -> "True" from the demonstrations. In the same perspective, the success case in Figure 1(a) can be attributed to learning desired patterns such as "Answer:" -> "True|False" in the demonstrations. Such patterns are apparently used as features in the autoregressive inference process by the model.

We take a feature-centric view to understand ICL, analyzing the key patterns in the input context that correlate with ICL behavior. The patterns we discussed above can be viewed as generalizations to repetition patterns (Holtzman et al., 2019; Fu et al., 2020) and self-reinforced patterns (Xu et al., 2022) which have been discussed in the literature. The 'self-reinforcement effect' describes a phenomenon where the model tends to perpetuate the generation of sentences that have frequently appeared in its context. These effects are regarded as harmful to text generation and previous work puts efforts

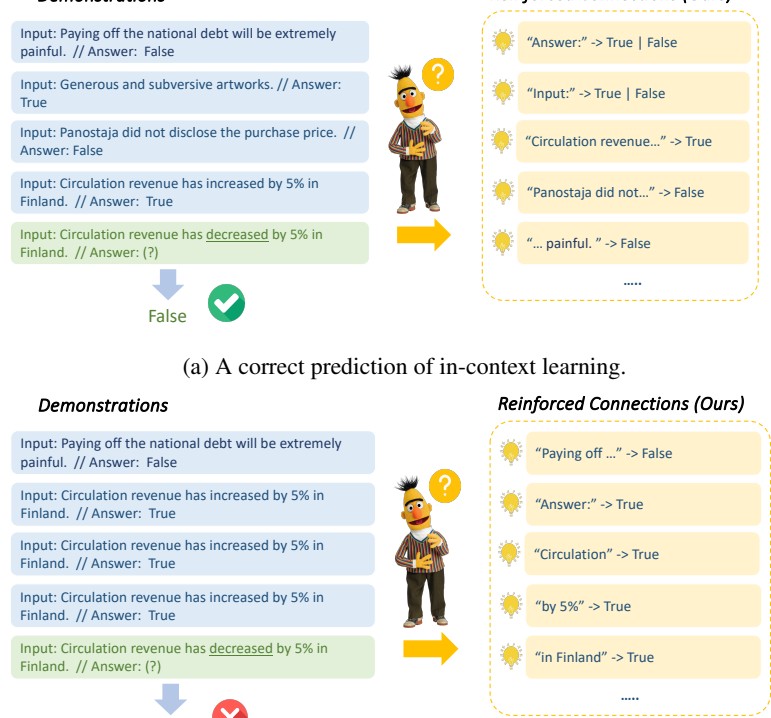

(a) A correct prediction of in-context learning.

(b) An in-correct prediction of in-context learning.

Figure 1: We showcase correct and incorrect predictions of in-context learning of LLaMA-65B. The shown task is to identify whether the given sentence presents a positive sentiment. We involve the token reinforced connections from demonstrations. In both cases, LLMs learn connections from the demonstrations and make decisions based on these connections. In the case of in-context learning, the model learns reliable connections and hopefully several of these connections result in the function of sentiment analysis. On the other hand, in repetitive demonstrations, the model gets stuck to spurious connections and misses the key information *'decreased'*, leading to a wrong prediction.

to mitigate it. However, they could give a unique view of the ICL behaviors from the angle of text generation.

We quantitatively investigate in-context learning from the perspective of surface patterns, illustrating the inherent correlation among surface patterns, self-reinforcement, and ICL. First, we study the roles of self-reinforce patterns as surface features that guide text generation. We empirically establish the existence of the *token co-occurrence reinforcement*, where the connection between any two tokens gets reinforced with the number of contextual co-occurrences, a primary principle in learning surface-level patterns. We further delve into the reasons and inner workings causing token reinforcement, showing it as an inevitable result of the model's efforts to maximize the likelihood of the training corpus.

Given the existence and reasons behind such patterns, we scrutinize the beneficial and detrimental effects of these surface patterns on in-context learning. On the one hand, experiments on MMLU and GSM8K show that the reinforcement helps constrain the output space and format outputs to follow demonstrations like outputting 'Let's think step by step.'. On the other hand, experiments with non-informative connections and reordered answers in MMLU demonstrate that intentionally constructed connections make LLMs lean towards specific answers, revealing the risk of unintended, spurious connections. Our work not only reveals the intrinsic workings of in-context learning to some extent, providing a perspective not analyzed in the literature but also explains the underlying reasons for the failure of in-context learning[1].

---

[1] https://github.com/ElliottYan/understand-icl-from-repetition

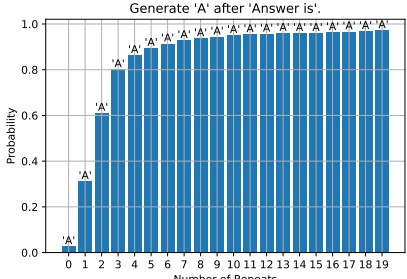 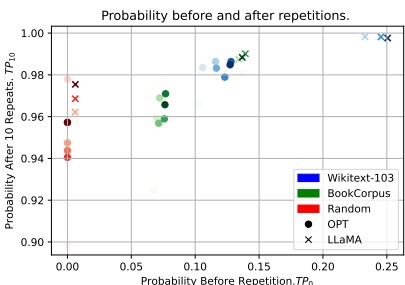

Figure 2: *Left:* **An example of the self-reinforcement effect.** We choose a normal sentence ('Answer is A'), repeat it several times, and present the probability of the token 'A'. The model used is LLaMA-7B. *Right:* **Sentence-level self-reinforcement over LLMs.** We plot all sizes of OPT and LLaMA with colors from light to dark. All sizes of LLaMA and OPT models demonstrate strong sentence-level self-reinforcement effects.

Our main contributions can be summarized as follows:

- We propose a novel perspective to understand ICL with repetitive text generations.
- We perform systematic analyses and empirically establish the existence of token reinforcement across various LLMs, alongside with the reason behind.
- We show that token reinforcement constrains output space and enables desired patterns for ICL, but is also responsible for spurious connections and possible failure of ICL.

## 2 ICL AND REPTITIONS

We first illustrate the background. In ICL, given a desired task $f$, we feed an LLM with $K$ input-output pairs $\{(x_k, y_k), n \in [1, K]\}$, where $y_k = f(x_k)$. Each input-output pair is formatted as a demonstration $d^k = (\mathbf{F_I}, x_k, \mathbf{F_O}, y_k)$. $\mathbf{F_I}$ and $\mathbf{F_O}$ denote formatting tokens for inputs and outputs, e.g., *'Input:'* and *'Answer:'*, and $x_k$ and $y_k$ can consist of several tokens.

A pretrained language model $\mathcal{M}$ predicts the output $y$ conditioned on the concatenation of demonstrations and the test query $x$,

$$\mathcal{P}_{\text{ICL}}(y|x, k) \coloneqq \mathcal{M}(y|(\mathbf{F_I}, x_1, \mathbf{F_O}, y_1, \cdots, \mathbf{F_I}, x_k, \mathbf{F_O}, y_k, \mathbf{F_I}, x, \mathbf{F_O})). \tag{1}$$

To understand the influence of surface repetitions as in Figure 1(b), we fist show a case of sentence repetition and plots the probability of the sentence as the number of repetition in the previous context increases (Figure 2). When we manually repeat the sentence 'Answer is A', the probability of generating 'A' after 'Answer is' gets boosted from 0.03 to almost 1.0.

The right part of Figure 2 demonstrates our quantitative study with two families of large language models, namely OPT (125M, 350M, 1.3B, 2.7B, 6.7B, 13B, 30B, Zhang et al. 2022) and LLaMA (7B, 13B, 30B, 65B, Touvron et al. 2023a). We plot the average sentence probability after 10 repeats for LLMs of varying sizes, arranged by light to dark colors. We use 1,000 sentences from each of the three different datasets – Wikitext-103 (Merity et al., 2016), BookCorpus (Zhu et al., 2015), and sequences of random words. More experimental details can be found in Appendix B. With 10 repeats, the probability of generating the sentence is significantly increased across all tested LLMs. LLMs amplify the occurrence of previously presented sentences, even *sequences of random tokens.*[2]

The above observations are related to the study of *self-reinforcement effect* (Xu et al., 2022) in literature. Formally, the conditional probability we model here is $\mathcal{P}_{\text{REP}}(w) \coloneqq \mathcal{M}(w|[s^1; s^2; \cdots; s^{n-1}; w_1 \cdots w_{i-1}])$, where $s$ is the repeating sentence, $n$ denotes the number of

---

[2]Xu et al. (2022) reported that sentences with a low initial probability — such as sentences composed of random tokens — have a smaller self-reinforcement effect. In our experiments, even sentences with random tokens (which initially have a near-zero probability) become reinforced to a probability nearing one. The difference may come from different model sizes (150M vs. maximum 65B) and pretrained corpus size (hundred millions of tokens vs. trillions of tokens).

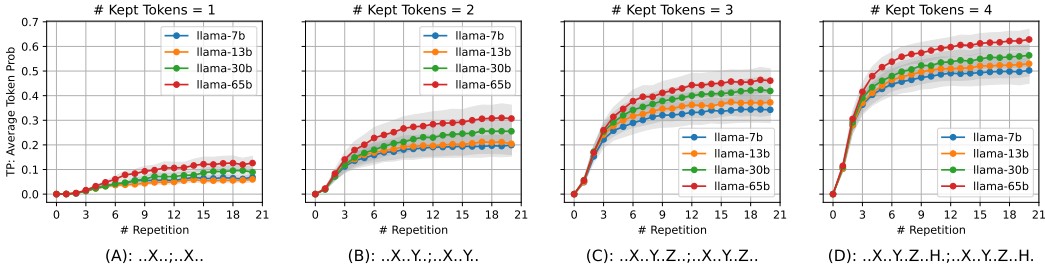

Figure 3: **Token co-occurrence reinforcement.** Even if only one token repeats in context, the self-reinforcement loop triggers. "..X..Y.." denotes 2 tokens are kept unchanged. The mean and variance are computed over 1,000 randomly generated samples.

occurrences, and $w_i$ is the $i$-th token in the sentence $s$. Previous research finds the self-reinforcement effect, where the probability of generating the sentence $s$ of length $L_s$ occurred $N$ times in context, $\text{TP}_N = \frac{1}{L_s} \sum_i \mathcal{P}_{\text{REP}}(w|w = w_i; n = N)$, almost monotonically increases with the number of $N$. While the generation of repetitive sentences above can be understood as the influence of a single surface feature, we investigate more sophisticated surface patterns, which can be causes to behaviors of in-context learning.

# 3 SELF-REINFORCED SURAFACE FEATURES FOR IN-CONTEXT LEARNING

Similar to previous definition of the demonstration, we set $s = [\mathbf{F_I}; x_1; \mathbf{F_O}; y_1]$ and have,

$$\mathcal{P}_{\text{REP}}(w|n = K) = \mathcal{M}(y|(\overbrace{\mathbf{F_I}, x_1, \mathbf{F_O}, y_1, \cdots, \mathbf{F_I}, x_1, \mathbf{F_O}, y_1}^{K \text{ times}}, \mathbf{F_I}, x_1, \mathbf{F_O})).$$
$$\mathcal{P}_{\text{ICL}}(y|x, k = K) = \mathcal{M}(y|(\mathbf{F_I}, x_1, \mathbf{F_O}, y_1, \cdots, \mathbf{F_I}, x_K, \mathbf{F_O}, y_K, \mathbf{F_I}, x, \mathbf{F_O})).$$

Comparing $\mathcal{P}_{\text{REP}}(w)$ to $\mathcal{P}_{\text{ICL}}(y)$, we find: (1) $\mathbf{F_I}$ and $\mathbf{F_O}$ are both repeated across demonstrations; (2) In repetitive generation, $x_1$ and $y_1$ are repeated, while in ICL, $x$ and $y$ are changing.

To investigate the correlation between surface patterns and the resulting answer $y$, we gradually expand self-reinforcement patterns toward in-context learning. We achieve this by introducing random perturbations to each demonstration, imitating the role of changing $x$ while leaving certain components, such as $\mathbf{F_O}$ and $y$, unchanged.

The experiments in this section are conducted over the dataset of randomly generated sentences as in Section 2 and with the four LLaMA models. The results on Wikitext-103 and BookCorpus, and results with OPT and various other LLMs can be found in Appendix D. For each experiment, we repeat the pattern 20 times and report the mean and standard deviation of the probabilities for the kept tokens.

## 3.1 ANY TWO TOKENS CAN CREATE A TOKEN REINFORCED LOOP

We assume that tokens are the base unit of reinforcement. By assessing the self-reinforcement effect among tokens, we understand the self-reinforcement effect from first principles.

Formally, given a sentence $s = (w_1, \cdots, w_{L_s})$ from a corpus $D$, we construct a binary mask sequence $\hat{m} = (\hat{m}_1, \hat{m}_2, \cdots, \hat{m}_{L_s})$, and we define a replacement operation $\mathbf{R}(w, \hat{m}) = \begin{cases} w_r, & \text{if } \hat{m} = 0 \\ w, & \text{if } \hat{m} = 1 \end{cases}$ that replaces $w$ with a randomly sampled token $w_r$ from the vocabulary if $\hat{m} = 0$ and keep it unchanged when $\hat{m} = 1$. Note that $w_r$ is independently sampled for each sentence and each position. As for mask sequence $\hat{m}$, we randomly sample positions to put the 0s and 1s of the mask sequence $\hat{m}$ and control the number of kept tokens. In this way, a sentence $s$ is transformed into $\hat{s}^n = (\mathbf{R}(w_1, \hat{m}_1), \cdots, \mathbf{R}(w_{L_s}, \hat{m}_{L_s}))$, where $\sum_{l \in [1, L_s]} \hat{m}_l = L_t$. Then, we report the average token probability $\hat{\text{TP}}_N$ as in the previous section. Suppose we have a sentence $s = (\text{Apple}, \text{there}, \text{is}, \text{red})$ and $m = (1, 0, 1, 1)$, and the target token $w = \text{red}$. Then, the demonstrations in this section will be like 'Apple there is red // Apple logo is red // Apple juice is red'.

Figure 4: **Successive and distant reinforcement.** The self-reinforcement effect is the strongest when two tokens are successive, i.e., distance=0. Otherwise, the reinforcement is smaller and appears insensitive to the distance. "..X.Y.." denotes the distance between two tokens is 1.

**Number of Tokens**   As depicted in Figure 3, even only one single token shared across demonstrations elicits self-reinforcement. The scenario with two tokens kept unchanged is particularly interested (Figure 3(B)), as it reveals a basic case that one token triggers the generation of the other one. We find that *the connection between any two tokens gets reinforced and the probability increases monotonically with the number of their contextual co-occurrences.* We refer to this base effect as the token co-occurrence reinforcement. When we increase the number of preserved tokens from 2 to 4, we observe a strengthening of the reinforcement effect. This is because each former token forms a reinforced connection with all the latter ones.

**Distance In-Between**   We further examine the role of distance in token reinforcement. This analysis is confined to two tokens. Figure 4 distinctly differentiates between successive tokens (distance= 0) and distant tokens (distance>= 1), which we term as *successive reinforcement* and *distant reinforcement*, respectively. The successive reinforcement significantly elevates the probability, from 0 to 0.4, with only several repetitions. Conversely, the distant reinforcement provides a moderate boost to the probability, from 0 to 0.2, and appears to be indifferent to the distance between tokens.

Across all experiments, we observe a marked increase in reinforcement as model size scales, especially in the distant token reinforcement. This consistent escalation suggests that larger LLMs are more capable of following complex patterns in in-context demonstrations, which is consistent with results from Wei et al. (2023). We provide more supporting evidence of token reinforcement in Appendix D.

The link we found between any two tokens forms the foundation of sentence-level self-reinforcement. Each token is strengthened by the ones before it. In ICL, common elements in demonstrations, such as pattern words, form connections with label words like "A, B, C, D".

## 3.2   TOKEN REINFORCEMENT IN THE PRE-TRAINING CORPUS

We find that token reinforcement could be a result of the model's effort to maximize the likelihood of the training data.

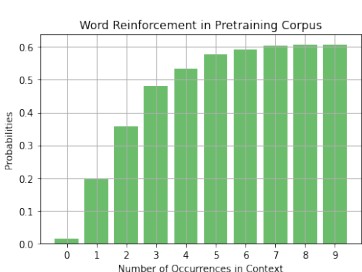

Figure 5: The probabilities of next occurrence after several occurrence observed in context.

Given a pre-training corpus $D_{train}$, the language modeling task over a sequence of tokens $\{w_1, w_2, \cdots, w_T\}$ with length $T$ is defined by $\max - \sum_i^T \log P(w_i|w_{<i})$. We compute the following empirical probabilities over the corpus:

$$P_n = \mathbb{E}_{w \in V}[\sum_i^T P(w_i = w|\text{Count}(w, w_{<i}) = n)], \forall n,$$

where $\text{Count}(\cdot, \cdot)$ counts the appearance of a word $w$ in the previous context $w_{<i}$, and $V$ is the vocabulary.

Intuitively, these probabilities reveal whether a particular word $w$ is likely to recur if there have already been $n$ instances of $w$ observed in the context, resonating with our analysis of token reinforcement.

**i-th Demonstration**

What are the mean and standard deviation of a binomial experiment that occurs with probability of success 0.76 and is repeated 150 times?

**A.** 114, 27.35

**B.** 100.5, 5.23

**C.** 114, 5.23

**D.** The mean is 114, but there is not enough information given to determine the standard deviation.

**Answer : D**

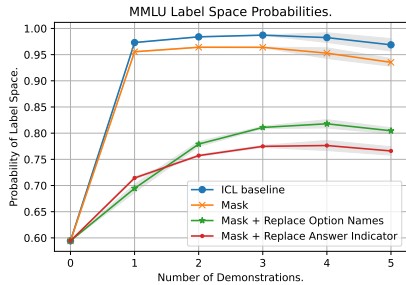

Figure 6: *Left:* **An example demonstration from MMLU's high school statistics dataset.** Colors indicate the parts to be masked. *Right:* **Probability of MMLU's label space.** We find that: (1) Masking question contents and answer contents of the demonstration does not influence directing the label space. (2) Both replacing the option names and the answer indicator significantly hurt the ability to constrain the label space. The gray shadow denotes the standard deviation across three runs.

We compute these probabilities over a commonly used pretraining corpus, wikipedia-english-2022[3], encompassing over 5.7B tokens, using the LLaMA tokenizer to first tokenize corpus and preprocess each context window with $T = 1024$ tokens.

Figure 5 demonstrates our results. We find that the trend of word probabilities accords to our results in Section 3.1. The more instances seen, the greater the probability of the same word recurring. Therefore, we infer that the token co-occurrence reinforcement stems from the LLMs' optimisation of the training likelihood. The LLMs manage to learn this inherent feature from the training data and generalize it to longer phrases and distant connections. This also elucidates the scaling with reinforcement, where larger models more effectively capture this feature from the training corpus.

### 3.3 UNDERSTANDING ICL VIA REINFORCEMENTS

Token reinforcement effect discussed in previous sections provides a new perspective to understand in-context learning. Consider the following example with three demonstrations [A,B,C,D ; A,b,C,D ; a,B,C,D ; A,b,C,(?)]. Our target is 'D'. In this example, several reinforced connections exist concurrently. 'A->D' is reinforced twice; 'b->D' is reinforced once; 'C->D' is reinforced three times. Here, all three reinforcements reach a consensus and predict 'D' in cooperation.

However, this ideal scenario is not always the case. Consider another example [A,B,C,D ; A,b,C,E ; a,B,C,F ; A,b,C,(?)]. At this time, 'A->D' is reinforced once; 'b->E' is reinforced once; 'a->F' is reinforced once and et cetera. These reinforcements create conflicts and compete against each other. Thus, instead of the traditional view of ICL as a mapping from input to output, we view ICL from a feature angle, as a combination of connection of tokens, even though some of them are highly reinforced and some of them are not. The next section shows how the reinforcements play a crucial role in in-context learning.

## 4 THE EFFECTS OF SURFACE PATTERNS TO IN-CONTEXT LEARNING

We quantitatively study how self-reinforced surface patterns lead to both beneficial functions and detrimental effects in ICL. The experiments in this section are conducted over MMLU (Hendrycks et al., 2021) and GSM8K (Cobbe et al., 2021). Due to limit of computing resources, we randomly sampled 20 samples for each of the 57 tasks of MMLU, resulting in a collection of 1140 test samples. The demonstrations are independently drawn for each of the test samples. All experiments are conducted across three random seeds. For further experimental details, see Appendix B.

### 4.1 BENEFICIAL EFFECTS

**Constraining Output Space.** An important advantage brought by the reinforced effect is that it helps constrain the output space — with several demonstrations, connections between formatting

---

[3] https://huggingface.co/datasets/olm/olm-wikipedia-20221220

tokens ('Input:' and 'Answer:') and label words in each demonstration ('ABCD') are reinforced, through distant and successive reinforcement, respectively. In this way, the LLM learns to predict either one of 'ABCD' as the final answer, instead of continuation sequences such as 'Oh, an interesting question. I hope I know the answer.'.

We verify this advantage with the MMLU dataset, which is widely used to evaluate the language understanding of real-world large language models. To isolate the effect of self-reinforcement, we construct masked demonstrations for analyses. An example of how we mask demonstrations is shown in the left part of Figure 6. Particularly, a demonstration in the MMLU dataset can be divided into five parts: question content, option name (e.g., 'A.'), option content (e.g., '114, 27.35'), answer indicator (e.g., 'Answer:') and final answer (e.g., 'D'). Based on token reinforcement, we hypothesize that the option names, i.e., "A.", "B.", reinforce outputting "A,B,C,D" via distant reinforcement. The answer indicator, i.e., "Answer: ", reinforces outputting within label space via successive reinforcement. To validate our hypothesis, we first mask all the questions and option contents in all demonstrations and keep the formatting words, final answer, and test query unchanged. Then, we further ablate option names and answer indicators by replacing them with semantically equivalent substitutes. More experimental details can be found in the Appendix B.

We use LLaMA-65B and plot the probability of choosing "A, B, C, D" as the predicted answer. The results are shown in the right of Figure 6. We find that masking the question and option contents, typically regarded as the most informative parts, does not influence the ability to direct the label space. Option names and answer indicators repeated several times in the demonstrations fulfill the job. Our conclusion is further solidified after ablating the option names and answer indicators individually. We see a huge decrease when replacing both option names and answer indicators, validating our hypothesis.

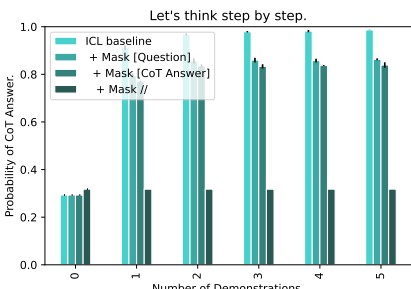

Figure 7: **Ability to follow patterns from demonstrations.**

**Learning to Follow Patterns.** Another distinctive feature of in-context learning is to follow the patterns of demonstrations. This is exemplified in techniques such as the Few-shot Chain-of-thought (CoT) prompting (Wei et al., 2022), frequently employed in reasoning tasks of LLMs like the GSM8K (Cobbe et al., 2021).

Here, we illustrate how the reinforced features in Section 3.1 affect the pattern following of ICL, by showing how LLMs follow the chain-of-thought demonstrations in the GSM8K high school math dataset. Each demonstration in the dataset follows the form "Question: [Question] // Let's think step by step. // [CoT Answer]". We demonstrate how models learn to say the CoT pattern, i.e., "Let's think step by step.". We further discuss the connection between surface patterns and [CoT Answer] in Appendix E.1.

Based on the findings in previous sections, we hypothesize that the common parts in the demonstrations teach the LLM to generate the CoT pattern. More specifically, 'Question:' builds a distant reinforcement, and the new liner '//' builds a successive reinforcement with the CoT pattern. We mask out each part in each demonstration *progressively* with random tokens to ablate the influences.

The probability of the CoT pattern is shown in Figure 7. After masking out "//", the probability gains obtained from demonstrations almost diminish, verifying our hypothesis of successive reinforcement. Another interesting finding is masking [Question] reduces the probability of generating the CoT pattern, indicating the [Question] part to some extent builds a connection with the CoT pattern. Since the questions in demonstrations are different but lie in the same group of grad math problems, there might be some common patterns among these questions.

## 4.2 DETRIMENTAL EFFECTS

Token reinforcement is not always helpful. In this section, we explore the detrimental consequences that arise from it. As shown in Section 3.1, distant and successive reinforcement are activated with even two random tokens. This could potentially lead to spurious patterns across demonstrations,

Table 1: *Top:* **Effect of Non-Informative Connections (NC).** The accuracy of D increases with the cost of A, B and C. *Bottom:* **Effect of Reordered Answers.** With more reordered demonstrations, the outputs are more leaned toward D. The delta values on the superscript denote the improvement compared with zero-shot scenarios. 'Avg. [A,B,C]' denotes the average accuracy of samples whose golden answer is A, B or C. 'D' denotes the accuracy of samples whose golden answer is D. "†": significantly different compared to its corresponding ICL baseline (p < 0.05).

| | *Non-informative Connections* | | | | | |
|---|---|---|---|---|---|---|
| # Demos | 0 | 1 | 2 | 3 | 4 | 5 |
| Avg. [A,B,C] | 63.39 | $63.74^{+0.35}$ | $64.56^{+1.17}$ | $64.17^{+0.78}$ | $64.44^{+1.05}$ | $64.37^{+0.97}$ |
| Avg. [A,B,C] w/ NC | 63.04 | $61.21^{-1.83†}$ | $62.57^{-0.47†}$ | $63.27^{+0.23†}$ | $63.00^{-0.04†}$ | $63.47^{+0.43}$ |
| D | 52.28 | $59.65^{+7.37}$ | $60.00^{+7.72}$ | $59.30^{+7.02}$ | $59.88^{+7.60}$ | $59.53^{+7.25}$ |
| D w/ NC | 49.47 | $63.63^{+14.15†}$ | $64.09^{+14.62†}$ | $63.51^{+14.04†}$ | $62.57^{+13.10†}$ | $61.64^{+12.16}$ |
| | *Reordered Answers* | | | | | |
| Avg. [A,B,C] | 63.39 | $63.74^{+0.35}$ | $64.56^{+1.17}$ | $64.17^{+0.78}$ | $64.44^{+1.05}$ | $64.37^{+0.97}$ |
| Avg. [A,B,C] w/ 50% RA | 63.39 | $64.56^{+1.17}$ | $64.44^{+1.05}$ | $64.05^{+0.66}$ | $63.94^{+0.55}$ | $63.86^{+0.47}$ |
| Avg. [A,B,C] w/ 75% RA | 63.39 | $64.52^{+1.13}$ | $62.65^{-0.74†}$ | $62.53^{-0.86†}$ | $61.95^{-1.44†}$ | $62.34^{-1.05†}$ |
| Avg. [A,B,C] w/ 100% RA | 63.39 | $64.80^{+1.40†}$ | $62.03^{-1.36†}$ | $61.17^{-2.22†}$ | $59.10^{-4.29†}$ | $58.05^{-5.34†}$ |
| D | 52.28 | $59.65^{+7.37}$ | $60.00^{+7.72}$ | $59.30^{+7.02}$ | $59.88^{+7.60}$ | $59.53^{+7.25}$ |
| D w/ 50% RA | 52.28 | $59.30^{+7.02}$ | $60.82^{+8.54}$ | $61.40^{+9.12}$ | $61.75^{+9.47}$ | $61.64^{+9.36}$ |
| D w/ 75% RA | 52.28 | $59.06^{+6.78}$ | $62.81^{+10.53}$ | $65.50^{+13.22†}$ | $67.02^{+14.74†}$ | $66.90^{+14.62†}$ |
| D w/ 100% RA | 52.28 | $58.71^{+6.43}$ | $66.20^{+13.92†}$ | $71.35^{+19.06†}$ | $75.67^{+23.39†}$ | $77.19^{+24.91†}$ |

which might be completely unforeseen by end users. We illustrate this point using two experiments where we manually construct spurious patterns in the MMLU dataset.

**Non-informative Connections.**   Our first approach is adding connections between a phrase and a certain choice. We append a reasonable but non-informative phrase such as *'Please kindly provide your answer.'* or *'Looking forward to your choice.'* right before the template *'Answer:'* each time the question's answer is 'D'. In testing, we also append the same phrase and check whether the outputs are navigated toward 'D'. By doing so, we construct a distant reinforcement loop from the non-informative phrase to the answer 'D'. We ensure the test set is balanced with equal numbers of questions having golden answers "A,B,C,D", and we report the accuracies at the top of Table 1.

We first see a gap even without demonstrations, where adding non-informative phrases lowers the accuracy of choices D. We further discuss the selection bias (Zheng et al., 2023) of different choices in the Appendix. Then, we see that the non-informative connection overcomes the selection bias and significantly elevates the accuracy choice D with a noticeable gap, in the cost of accuracy of A, B, and C. These results show the potential risk of manually injecting spurious connections and directing in-context learning toward unintended outcomes.

**Answer Indicator Connections.**   In our second experiment, we show that reinforcing the connection between 'Answer:' and a certain choice, e.g., D, navigates the outputs. To this end, we randomly replace $r$ percent of demonstrated answers with D. Simultaneously, we exchange the option contents of the original golden answer and D, to keep the demonstrations valid. In this way, we gradually reinforce the connection between 'Answer:' and 'D', with successive reinforcement.

The results are presented at the bottom of Table 1. Our baseline is 25%, where the demonstrations are balanced. With the increased ratio of answer D in demonstrations, the accuracy of D is largely improved, from 0.52 to 0.78, while the accuracy of A, B, and C decreases from 0.63 to 0.58. Our findings corroborate with An et al. (2023), where they show how diversity affects the performance of in-context learning. Our findings demonstrate how unbalanced demonstrations bring an unfair advantage for certain outputs.

**Discussion**   (1) Reinforcements can be the underlying reason for ICL, but also causes its vulnerability. (2) Our observations guide how to build demonstrations to maximize ICL effectiveness! The demonstrations should be both balanced for all possible output labels, and kept concise enough without introducing any unnecessary reinforcement.

## 5    RELATED WORK

**Explaining In-Context Learning.** A range of contributions has deepened our understanding of In-Context Learning (ICL). Chan et al. (2022) and Xie et al. (2022) explore the emergence of ICL from the perspective of training data and Bayesian inference, respectively. Implicit learning of ICL over demonstrations is further highlighted by Garg et al. (2023), Li et al. (2023), and Hahn & Goyal (2023) theoretically show that performance of ICL can be represented by a complexity that repetition structures can be represented by a small PCFG tree. The similarity between gradient descent learner and in-context learner is demonstrated by Akyürek et al. (2023); von Oswald et al. (2023), while Dai et al. (2023) explain language models as meta-optimizers and likens ICL to implicit finetuning. Our work differs from this line of work with a novel perspective via repetitions and reinforced features, and our findings could potentially explain the mechanism of how LLMs achieve implicit learning. For instance, the step by step reinforcement across demonstrations intuitively resembles the gradient descent process of ICL described in previous work. Olsson et al. (2022) introduce induction heads of copying patterns and provide evidence for their relationship with ICL. Differently, our work investigates the LLM as a whole (Anderson, 1972), studies sophisticated patterns,  views ICL as a combination of reinforcements, and scrutinizes both the benefits and drawbacks of reinforcements.

**Analyzing In-Context Learning.** Several studies have analyzed ICL properties. Min et al. (2022); Yoo et al. (2022) identify and discuss key factors that influence ICL capability such as input-label mappings.  Wei et al. (2023) proposes that learning input-label mapping is an emergent ability. Factors such as structural similarity, diversity, simplicity (An et al., 2023) , and order or embedding distribution (Lu et al., 2022; Liu et al., 2022) are also investigated.   Pan et al. (2023) partitions ICL ability into task recognition and task learning, observing different phenomena with varying model sizes. Lastly, Si et al. (2023) unveils the presence of inductive biases in ICL by designing underspecified demonstrations.

Our findings corroborate with multiple previous analyses of in-context learning. For example, the scaling for distant reinforcement echoes Wei et al. (2023); Pan et al. (2023)'s findings of different phenomena when varying model sizes. The importance of demonstration ordering and diversity in An et al. (2023); Lu et al. (2022) can be explained by avoiding spurious connections.

**Repetitive Generation and Self-Reinforcement Effect.** Repetition is a notorious issue in neural text generation, affecting tasks like open-ended and directed generation (Holtzman et al., 2019; Welleck et al., 2019; Lin et al., 2021; See et al., 2017; Liu & Lapata, 2019). Maximization-based decoding strategies lead to bland, consecutive repetitions at word, phrase, and sentence levels (Holtzman et al., 2019; Welleck et al., 2019; Li et al., 2016; Karpathy & Fei-Fei, 2015; Guan et al., 2021). Despite advancements in large-scale pre-training with Transformer architecture (Vaswani et al., 2017; Radford et al., 2019; Lewis et al., 2020), unexpected sentence-level repetitions persist (Radford et al., 2019; Brown et al., 2020; Fu et al., 2020).

The repetition issue is puzzling given the lack of repetitive sentences in the training data. A series of studies investigate the cause, with both from the theoretical perspective (Fu et al., 2020) and empirical findings (Holtzman et al., 2019). Recently, Xu et al. (2022) proposes the *self-reinforcement effect*, suggesting a repetitive loop when combined with maximization decoding. Our study extends the effect to large language models and token reinforcement, explains the reasons, and bridges the excellent ability of in-context learning to this notorious issue.

## 6    CONCLUSION

We have taken a novel feature-centric approach to understanding in-context learning, by exploring its relationship with repetitive generation. We have identified a key mechanism, the token reinforcement loop, where any two tokens can form a strong connection through multiple co-occurrences. We delve into the reasons and inner-working of token reinforcement, demonstrating it to be an inevitable result of maximizing likelihood. Based on our findings, we view in-context learning as a combination of token reinforcements with different level of strength instead of input-label mapping. Furthermore, we conduct experiments to demonstrate that token reinforcement plays a crucial role in shaping the output space and following patterns in in-context learning. We also illustrate through various studies how token reinforcement leads to spurious connections in in-context learning, highlighting the role of in-context learning as a double-edged sword, where informed demonstrations can maximize ICL effect.

## ACKNOWLEDGEMENT

This publication has emanated from research conducted with the financial support of both the National Natural Science Foundation of China Key Program under Grant Number 62336006 and the Pioneer and "Leading Goose" R&D Program of Zhejiang under Grant Number 2022SDXHDX0003.

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

# Appendix

## Table of Contents

## A  LIMITATIONS

While this study provides some insightful findings in the field of in-context learning, there are some limitations that should be noted. First, the experiments in this work constrain themselves to repeating surface patterns. More sophisticated patterns are not discussed. Second, our work mainly focuses on revealing the token co-occurrence reinforcement and understanding its influence on in-context learning. Its detrimental influences on in-context learning suggest that resolving the spurious connections would be helpful to either chain-of-thought or in-context learning.

## B  EXPERIMENTAL DETAILS

### B.1  DATASET DESCRIPTIONS

In this paper, we mainly use the following five datasets, and we introduce each of them and describe our preprocess of these datasets individually. We present cases from each dataset to demonstrate their characteristics in Table 2.

**Wikitext-103**   The Wikitext-103 dataset, introduced by Merity et al. (2016)[4], is a language modeling dataset that contains a collection of over 100 million tokens extracted from the set of verified Good and Featured articles on Wikipedia. We use a randomly sampled collection of 1000 sentences provided by Xu et al. (2022)[5]. The provided version is pre-tokenized to words and we use the moses detokenizer[6] to restore the untokenized version for compatibility with the tokenizer of transformers.

---

[4]`https://blog.salesforceairesearch.com/the-wikitext-long-term-dependency-language-modeling-`
[5]`https://github.com/Jxu-Thu/DITTO/blob/main/data/wiki_sentences.txt`
[6]`https://github.com/moses-smt/mosesdecoder/blob/master/scripts/`
`tokenizer/detokenizer.perl`

Table 2: Datasets and Cases

| Dataset | Number of Cases | Examples |
| --- | --- | --- |
| Wikitext-103 | 1000 | *The Bintulu Government Secondary School was built in 1964.* |
| BookCorpus | 1000 | *"A little, "he admitted.* |
| Random | 1000 | *Green Ou incarcer hijab ura na Edmonton regardless iken Mayor* |
| GSM8K | 1000 | *Question: Janet's ducks lay 16 eggs per day. She eats three for breakfast every morning and bakes muffins for her friends every day with four. She sells the remainder at the farmers' market daily for $2 per fresh duck egg. How much in dollars does she make every day at the farmers' market?*

*Let's think step by step.*

*Janet sells 16 - 3 - 4 = «16-3-4=9»9 duck eggs a day. She makes 9 * 2 = \$«9*2=18»18 every day at the farmer's market. ### 18* |
| MMLU | 1140 | *As more lamps are connected in parallel in a circuit, the current in the power source*
*A. increases*
*B. decreases*
*C. remains the same*
*D. Not enough information to say*
*Answer: A* |

Table 3: Semantically equivalent substitutes.

| Category | Original | Substitutes |
|---|---|---|
| Option Names | A.; B.; C.; D. | *I.; II.; III.; IV.* 
 *E.; F.; G.; H.* 
 *1.; 2.; 3.; 4.* 
 *(a).; (b).; (c).; (d).* |
| Answer Indicator | Answer: | *Solution* 
 *Reply* 
 *Response* 
 *Result* 
 *Choice* |

**BookCorpus**    BookCorpus (Zhu et al., 2015) is originally introduced to align the books to their movie releases in order to provide rich descriptive explanations for visual content. It contains 74M sentences from various sources of books, and we randomly sample 1000 sentences and also detokenize with moses.

**Random Generated Sentences**    Here, we follow Xu et al. (2022) and construct 1000 randomly generated sentences. For each sentence, we first sample a random length from 5 tokens to 10 tokens, and then sample the tokens from the whole vocabulary. All random sentences are generated by uniformly sampled from the vocabulary. All the symbols are equally likely to appear, except the special tokens.

**GSM8K**    GSM8K (Cobbe et al., 2021) is a dataset of high-quality grade school math problems created by human problem writers. The dataset contains 7500 training problems and 1000 test problems. All the problems are answered with between 2 and 8 steps to solve. It is a frequently used benchmark to evaluate the reasoning ability of large language models (Touvron et al., 2023a; Chowdhery et al., 2022). To analyze the chain-of-thought (Wei et al., 2022) effect of LLMs, we add "Let's think step by step" followed by the question and right before the answer.

**MMLU**    The MMLU (Hendrycks et al., 2021) dataset is another commonly used benchmark to evaluate the knowledge of large language models. It covers 57 subjects across STEM, the humanities, the social sciences, and more. It ranges in difficulty from an elementary level to an advanced professional level, and it tests both world knowledge and problem-solving ability. We randomly sample 20 test cases from each task, resulting in 1140 test queries in total. In the 'undesired' section, we uniformly redistribute the answers and options to isolate the selection bias of LLMs.

## B.2    EXPERIMENTAL DETAILS

**Random Substitutes**    Throughout the paper, we adopt random substitutes to isolate the effects of tokens, specific formats, and other components. The random substitutions are conducted in the following manner. To avoid the effect of different sequence lengths, we first tokenize the original sentence or demonstration using the Transformers tokenizer. Then, we replace the part to substitute with random tokens from the corresponding vocabulary. With the substitutes, we exclude the special tokens to ensure that all random tokens are valid.

**Self-Reinforcement**    For each experiment, we report the averaged results across three runs with different seeds. We mainly conduct our experiments over two model families, LLaMA and OPT. For LLaMA models, we use 4 models sized in [7b, 13b, 30b, 65b]. For OPT models, we use 7 models with sizes ranging in [125m, 350m, 1.3b, 2.7b, 6.7b, 13b, and 30b]. Each sentence is repeated 20 times in our experiments. Following Xu et al. (2022), we randomly concatenate a prefix before all repeated sentences.

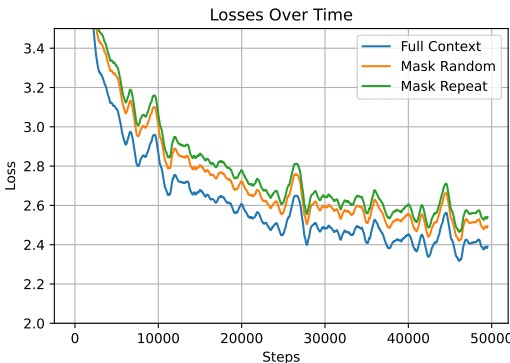

Figure 8: **Pretraining losses when masking or not masking the repetitive features.**

**In-Context Learning**    In the MMLU experiments, we replace option names and answer indicators to study the importance of token reinforcement in directing output space. Specifically, for each replacement, we choose semantically equivalent substitutes from a pool and randomly replace the original option name/answer indicator with the chosen substitute. In this way, we break the token reinforcement from the demonstrations. We put the pool of our replacement in Table 3.

**Significance test**    We conduct the significance test using paired t-tests, where we randomly split the test set into 5 folds and compute the significance over accuracies on these 5 subsets. In Table 1, we compute the significance levels for [A, B, C] and D separately.

## C  FURTHER INVESTIGATION OF REASONS AND INNER-WORKINGS FOR TOKEN CO-OCCURRENCE REINFORCEMENT

In Section 3.2, we investigate the reason of token reinforcement as it is inherently embedded within the pretraining corpus. Here, we further investigate token reinforcement with two more experiments, from the learning process and attention mechanism, respectively.

**Leveraging repetitive features helps maximizing likelihood.**    Our next experiment shows that the utlization of repetitive features is a natural choice of LLMs as it helps in the pre-training stage.

To illustrate this, we focus on a conditional language modeling task that predicts tokens based on a given prompt. Formally, we model the probability $P(w_{T_p:T}|w_{<T_p})$, where $T$ and $T_p$ are the total context length and the number of tokens to condition on, respectively.

We then consider three types of prompts. The first one serves as our baseline prompt with $w_{<T_p}$. In the second type, we mask the tokens in the prompt that appear in $w_{T_p:T}$. For the third type, we mask the same quantity of tokens as in the second type, but we select the tokens randomly.

For each type of prompts, we pretrain a language model. The base architecture of our model is the same as LLaMA. Due to limitation of computational cost, we make several changes in hyperparameters for a smaller size. We set the hidden size to 1024 and the FFN size to 4096, and we incorporate 12 layers. The tokenizer is the same as LLaMA and the vocabulary size is 32000. This configuration results in a model with 267M trainable parameters. The dataset we use is the wikipedia-english-2022, and the code base we use is LLaMA-Factory (hiyouga, 2023). We pretrain each model for 50k steps, with a total batch size of 160 sentences per step. Each sentence contains 1024 tokens. Prompt length $T_p$ is 0.75 * 1024 = 768 tokens. We run our experiments on 4 NVIDIA A100 GPUs, and each experiment takes about 30 hours to finish.

Our results are shown in Figure 8. Masking repetitive features lead to a worse converged loss compared with masking random tokens and no masking. It indicates that repetitive feature is favorable in optimizing training loss, and thus making it natural for language models to use repetitive features.

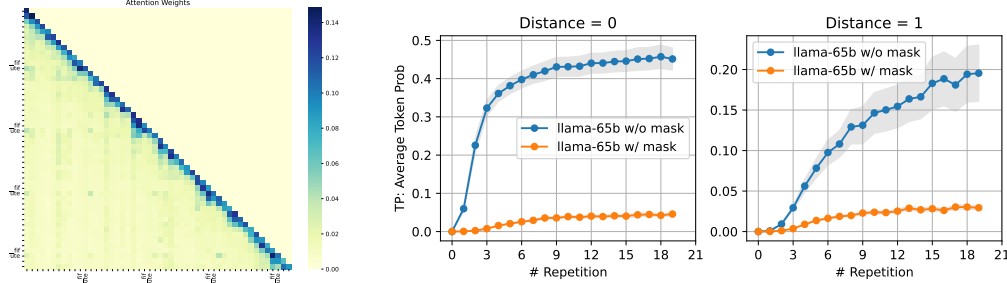

Figure 9: *Left:* **Attention example of successive token reinforcement.** '_fif' and 'ute' are the two kept tokens randomly chosen. *Right:* **Ablation study of attending preceding token.** The results are based on LLaMA-65B.

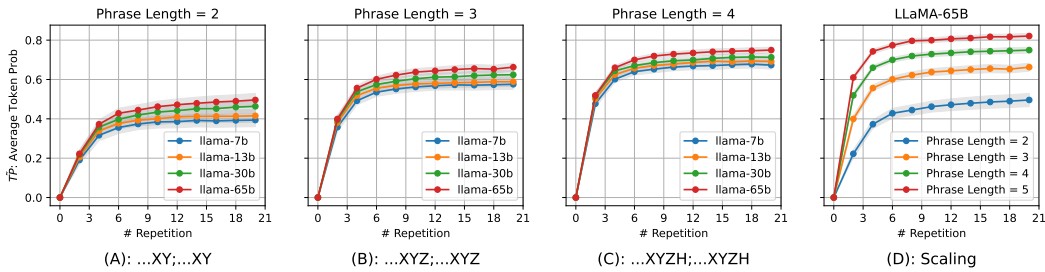

Figure 10: **Phrase co-occurrence reinforcement.** When disrupting the reinforcement loop with random prefixs, the self-reinforcement effect still retains. (A)-(C) plot the scaling of LLaMA models with a certain phrase length (e.g., "XY" denotes length of 2). (D) plots LLaMA-65B's probability varying phrase length. The gray shadow denotes the standard deviation across 1,000 samples.

**Attending the preceding token is responsible for reinforcement.** Another important question is what inner-working of LLMs is responsible for token reinforcement? We start our investigation with a visualization of attention weights. An example is shown on the left of Figure 9. In this example, each token $w_i$ attends its preceding tokens $w_{i-1}$ with relatively high attention weights. Thus, our intuition is that attending the preceding token is the key to reinforcing the connection.

To validate this hypothesis, we reproduce the successive reinforcement experiment in Section 3.1, with an attention mask preventing each token from attending its preceding token. Our results are shown in the right of Figure 9. We observe that upon masking the adjacent token, the strength of reinforcement is reduced by a factor of ten. Hence, the reinforcement is not simply realized by attending to similar tokens. Rather, the information is propagated through a process wherein each token iteratively attends to its adjacent token.

# D SUPPORTING EXPERIMENTS OF TOKEN CO-OCCURRENCE REINFORCEMENT

In this section, we provide more experimental evidence related to the token co-occurrence reinforcement. The experimental settings in this section follow the same setting as in Section 3.1, except for models to use, datasets to use, and the choices of tokens.

## D.1 PHRASE CO-OCCURRENCE REINFORCEMENT

We first construct consecutive tokens to create phrase reinforcement in this section. Specifically, we Without loss of generality, we place the phrase at the end of each demonstration.

We construct a binary mask sequence $m = (\overbrace{0, \cdots, 0}^{L_s - L_p}, \overbrace{1, \cdots, 1}^{L_p})$, where $L_p$ is the length of the kept phrase. Then, we can define a randomly perturbed sentence $\tilde{s} =$

$(\mathbf{R}(w_1, m_1), \cdots, \mathbf{R}(w_{L_s}, m_{L_s}))$. Effectively, we keep a phrase of length $L_p$ unchanged for each sentence, and replace other tokens with random tokens. We compute $\mathcal{P}_{\text{REP-P}}(w) = \mathcal{M}(w|[\tilde{s}^1; \tilde{s}^2; \cdots; \tilde{s}^{n-1}; \mathbf{R}(w_1, m_1), \cdots, \mathbf{R}(w_{i-1}, m_{i-1})])$ and the average token probability that only considers the kept tokens $\tilde{\text{TP}}_N = \frac{1}{L_p} \sum_{i=L_s-L_p+1}^{L_s} \mathcal{P}_{\text{REP-P}}(w|w = w_i, n = N) \times m_i$. As a concrete example, we take a sentence $s = (\text{Apple}, \text{is}, \text{red})$ and $m = (0, 1, 1)$, and the target token $w = \text{red}$. Then, the demonstrations discussed will be like 'Apple is red // Orange is red // Banana is red'. The pattern we kept unchanged is in color red. In the context of in-context learning, this pattern corresponds to demonstrations like '...Answer: D' or '...Response: Positive'.

Figure 10 depicts our experimental results for repeated and discontinuous phrases. We use randomly generated sentences here. We see that disrupting the sentence-level loop with random prefixes does not break the self-reinforcement effect. The effect persists across various LLaMA models. In addition, we observe a stronger reinforcement when increasing the phrase length. More tokens are kept unchanged, higher probability the model assigns to the phrase. Particularly, Figure 10(D) serves as an ablation study, as each line progressively adds one more unchanged token. A repeating phrase of '...XYZ' gets reinforced more compared to the phrase '...XY', indicating that each repetitive token bolsters its subsequent tokens.

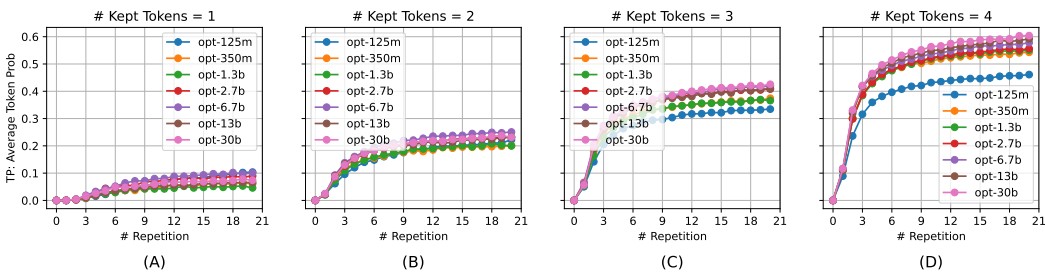

Figure 11: **Token co-occurrence reinforcement of OPT models.**

## D.2 TOKEN REINFORCEMENT OF OPT MODELS

In Figure 11, we plot the token reinforcement of all 7 OPT models. The results are consistent with our results of LLaMA in the main context, validating the generality of token reinforcement across different families of large language models.

## D.3 TOKEN REINFORCEMENT OF OTHER LLMS

We present token reinforcement on LLaMA-2 (Touvron et al., 2023b)(7B, 13B, 70B), Mistral (Jiang et al., 2023)(7B), GPT-J (Wang & Komatsuzaki, 2021)(6B). Figure 12 demonstrates that all these LLMs exhibit similar token reinforcement effect, with various strength. These results consolidate our findings.

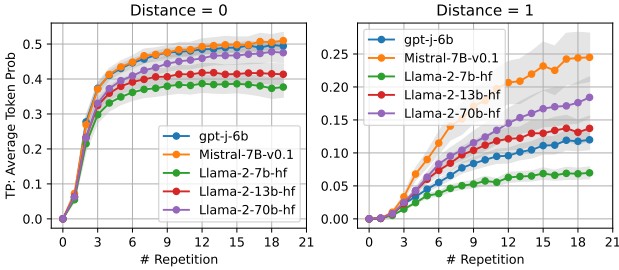

Figure 12: **Token reinforcement on Various LLMs.**

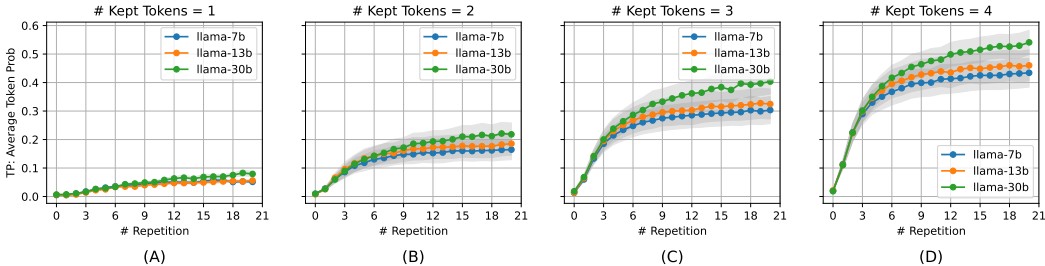

Figure 13: **Token co-occurrence reinforcement on Wikitext-103.**

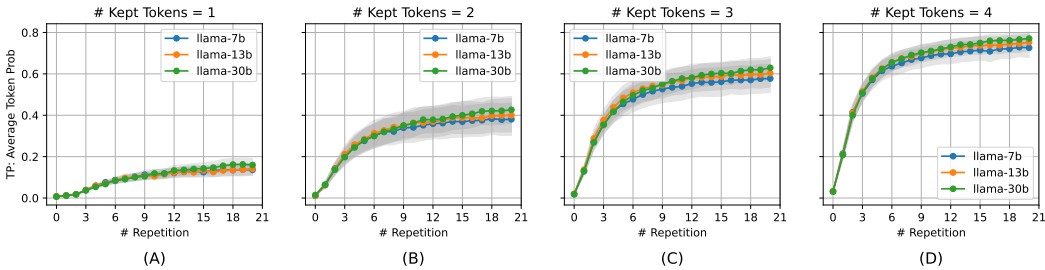

Figure 14: **Token co-occurrence reinforcement on BookCorpus.**

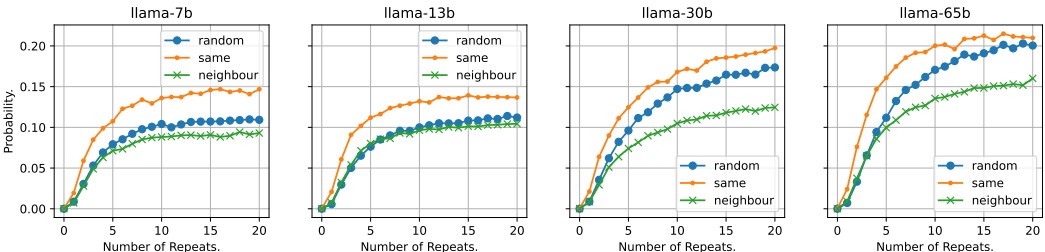

Figure 15: **Token reinforcement against semantic relationships.**

## D.4    TOKEN REINFORCEMENT ON OTHER DATASETS

Here, we plot the token reinforcement on Wikitext-103 and BookCorpus. As we can see, the probability of reinforced tokens is quite different in the two datasets. In BookCorpus, with 4 kept tokens, the probability can be boosted to about 0.8, whereas in Wikitext-103, the probability can only reach about 0.5. Note that compared to the results on randomly generated sentences, tokens in these two datasets are more likely to co-occur in the pre-training data. This indicates the semantic relationship between tokens affects how token reinforcement performs.

## D.5    SEMANTIC RELATIONSHIPS OF TOKEN REINFORCEMENT

We further conduct experiments to investigate how the semantic relationship between two tokens affects the token co-occurrence reinforcement. We choose three relationships: (1) Random two tokens. (2) The same two tokens. (3) Two tokens that are similar in the embedding space.

Figure 15 plots our results for different LLaMA models. We observe clear gaps among different semantic relationships. Two tokens that are the same can build a strong connection stronger than two tokens that are similar in the embedding space. Further investigating the reasons behind this is interesting and may unravel the internal biases of large language models. We leave it as the future work.

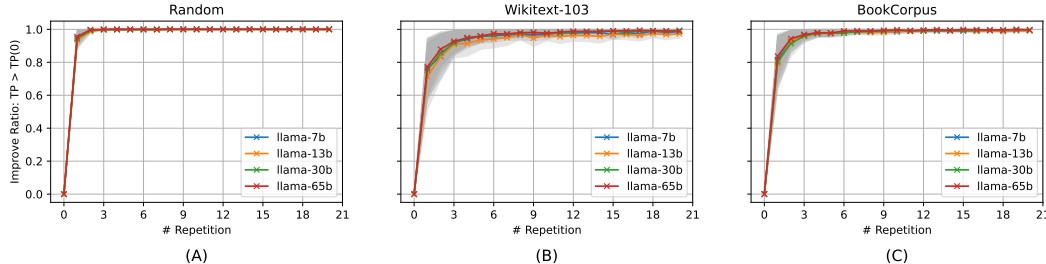

Figure 16: **Improved Ratio on three datasets.**

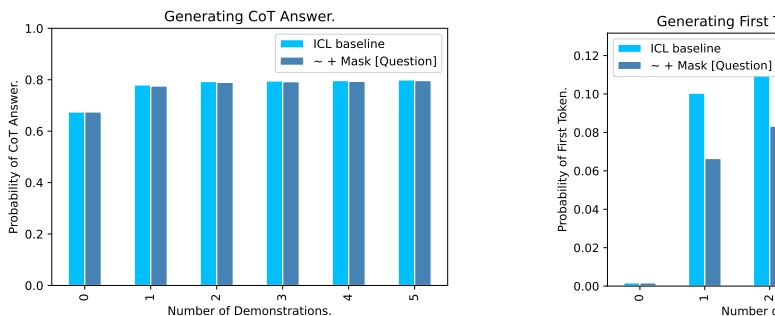

Figure 17: *Left:* **The probability of generating the CoT answer.** *Right:* **The probability of generating the first token in CoT answer.**

### D.6 IMPROVED RATIO OF TOKEN REINFORCEMENT

In this section, we solidify our findings with the ratio of improved token probability. Formally, the improved ratio (IR; Xu et al. (2022)) is defined as follows:

$$\hat{\mathbf{IR}} = \mathbf{T}\hat{\mathbf{P}}_N > \mathbf{T}\hat{\mathbf{P}}_0. \tag{2}$$

Our metric of improved ratio is defined over the setting in Section 3.1. Figure 16 plots our results of IR on three datasets. We only consider the case of two tokens.

As we can see, the improved ratios for all three datasets quickly reach almost 1.0 with only several repetitions, indicating token reinforcement exists in most of the cases. In addition, we observe that IRs in Wikitext-103 and BookCorpus have larger variances than those in randomly generated sentences, because tokens in Wikitext-103 and BookCorpus are more likely to co-occur in the same context and thus get larger probabilities without reinforcement.

## E OTHER EXPERIMENTS ON DETRIMENTAL EFFECTS

### E.1 GENERATION OF COT ANSWER

In this section, we study how the CoT answer is generated with respect to our discovered surface patterns. We study the problem on the GSM8K dataset. Recall the demonstrations in GMS8K, i.e., "Question: [Question] // Let's think step by step. // [CoT Answer]". We plot the probability of [CoT Answer] in Figure 17. We would expect the [Question] to have a huge effect on generatin the CoT answer. Thus, we involve the probability with random substitution of the [Question] part. Interestingly, we find that even without the [Question] part, the probability of [CoT Answer] still increases with demonstrations, suggesting the patterns play a crucial role in learning how to generate the CoT answers.

So what does the [Question] part do in in-context learning of GSM8K? We further plot the probability of the first token of [CoT Answer]. The probability of the first token significantly decreases when we mask out [Question]. Hence, even though the [Question] part does not affect much the probability of [CoT Answer], it substantiates the generation of [CoT Answer] at the very beginning.

E.2   SELECTION BIAS OF LLMS

In experiments illustrated in table 1, we observe a selection bias of LLaMA-65B. We show the results for each class in Table 4. Note that we randomly permute the test set to make sure all questions are balanced with [A,B,C,D].

First, we see that the zero-shot performances in different class are quite different. Class 'A' gets an accuracy as high as 71.58% and class 'D' only gets 52.28%. Second, with more demonstrations, the overall accuracy is improved, from 60.61% to 63.16%, demonstrating the effectiveness of ICL. However, the accuracy of class 'A' largely decreases, while the accuracies for 'B', 'C', and 'D' all increase. The above findings indicate that LLaMA-65B has a selection bias in both zero-shot and few-shot scenarios.

Table 4: **Accuracies for each class on our balanced MMLU.** We use the LLaMA-65B and vary the number of demonstrations from 0 to 5.

| Class | 0 | 1 | 2 | 3 | 4 | 5 |
|-------|-------|-------|-------|-------|-------|-------|
| A | 71.58 | 57.31 | 58.25 | 57.31 | 56.96 | 54.74 |
| B | 59.65 | 66.90 | 69.94 | 70.41 | 69.59 | 70.76 |
| C | 58.95 | 67.02 | 65.50 | 64.80 | 66.78 | 67.60 |
| D | 52.28 | 59.65 | 60.00 | 59.30 | 59.88 | 59.53 |
| Avg. | 60.61 | 62.72 | 63.42 | 62.95 | 63.30 | 63.16 |

