# OpenReview forum: "Understanding In-Context Learning from Repetitions"
_ICLR.cc/2024/Conference — ICLR 2024 poster_

### Official Review · Reviewer_Ydk8 · 2023-11-01

**Soundness:** 3 good
**Presentation:** 3 good
**Contribution:** 3 good
**Rating:** 6
**Confidence:** 3

**Summary:**

The paper examines the phenomenon of token co-occurrence reinforcement, whereby tokens occurring in the context have a higher chance of being predicted by an LM.
An interesting consequence is that In-Context Learning (ICL) can go wrong when a label that is different from the target label is repeated often in the in-context examples, potentially helping explain some failure modes of ICL.

It first shows that self-reinforcement occurs with randomly constructed sequences of words. The longer the repeated part, the stronger the reinforcement effect (Figure 3).
It then shows that even discontinuous repeated subsequences give rise to the reinforcement effect.
Section 4 studies the effect of the reinforcement effect on ICL. The effect can help constrain the output space and the desired pattern, but it can also lead to incorrect results.

**Strengths:**

- systematic study using openly available LLM family

- interesting results about the factors impacting the success and failure of ICL. For instance, in Figure 7, the authors seem to show that that in a specific CoT prompting setup (GSM8K), replacing the questions and CoT answers in the demonstrations with random tokens does not hurt ICL performance nearly as much as replacing a separator with random tokens.

**Weaknesses:**

- The paper remains somewhat unclear regarding the overall contributions and implications. The paper argues that the results helps understand both limitations of ICL and the inner workings of ICL.
 To the extent that the paper aims to illuminate the "inner workings" of ICL, the contribution is left somewhat unclear. Do the authors argue that co-occurrence reinforcement is implicated in LMs' ability to pick up input-label mappings? And what "inner workings" are responsible for the co-occurrence effect? A range of recent research discussed in Section 5 aims to explain how ICL works, and observations on the self-reinforcement effect could help shed light -- and indeed the paper hints at this repeatedly (e.g., end of page 4, "In the context of ICL, this pattern corresponds to demonstrations like" -- it seems that the experiment is understood as some kind of simple prompt-like structure, but this idea and its implications are then not made explicit).

**Questions:**

- Clarity: Section 3.1 uses the term "sentence" for the examples, but in the experiment they are "randomly generated". How are they sampled -- just as random sequences of symbols sampled i.i.d. from the vocabulary? Are all symbols in the vocabulary equally like to appear? The same question about how masked subsequences are resampled also applies to the other experiments.

- Section 4.1, "Learning to follow patterns" -- when masking the Question, does this mean that the questions in the demonstrations are masked (i.e., replaced with random word sequences) but the question in the final element of the prompt (the one to respond to) is not masked?

---

> ### Author Response · Authors · 2023-11-21
> **Author Response to Reviewer Ydk8**
>
> Thanks for your time and insightful reviews.
>
> **Q: “unclear contributions”**
>
> A: Thanks for your advice. Here, we restate our contributions as follows:
>
> - We propose a novel perspective to understand icl with repetitive text generations, where the key effect causing repetitions appears to be an important feature in ICL.
> - We empirically establish the existence of token reinforcement across various LLMs, alongside with the inevitable reason behind learning process and inner workings responsible for it.
> - We show that token reinforcement plays crucial role in ICL for both beneficial and detrimental effects. It constrains output space and desired patterns, but it is also responsible for spurious connections and possible failure of ICL. Our findings provide a novel research basis for analyze the mechanism of in-context learning.
>
> **Q: “what inner-working are responsible for token reinforcement”**
>
> A: We conduct more analysis over the attention weights and find that in our reinforcement experiments, the LLMs tend to attend the preceding adjacent token with high attention weights. Thus, we hypothesize that the attention to adjacent token is crucial for token reinforcement. To validate our hypothesis, we mask the each token’s attention to its preceding adjacent token and find the strength of token reinforcement is reduced by ~10 times.
>
> Hence, the reinforcement is not simply realized by attending to similar tokens. Rather, the information is propagated through a process wherein each token iteratively attends to its adjacent token.
>
> For more details about our experiments, we refer you to Appendix C.
>
> **Q: “clearer implications”**
>
> A: Thanks for your advice about implications and it helps improve our paper! Instead of the traditional thinking of ICL as a mapping from input to label, in this paper, we think of ICL as a combination of reinforcement of tokens, even though some of them only appear once and not get strongly reinforced. Currently, in the last paragraph of Section 3 and the conclusion of our revised paper, we highlight this implication, and we are making more efforts to make the paper clearer.
>
> **Q1 “random sentences”:**
>
> A: Yes. All random sentences are generated by uniformly sampled from the vocabulary. All the symbols are equally likely to appear, except the special tokens. We add the description to Appendix B.1.
>
> **Q2 “mask question”:**
>
> A: Yes. In this experiment, we are trying to test which reinforcement is helpful for following patterns. Thus, we do not touch the test query, and only modify the demonstrations.
>
> We add the above mentioned experiments to the paper and promise to update the main content accordingly. We hope these results resolve your concern and will be really appreciated if you re-consider the score of our paper.

---

> > ### Comment · Reviewer_Ydk8 · 2023-11-21
> >
> > I’ve increased the contribution score to 3.

---

> > > ### Author Response · Authors · 2023-11-22
> > >
> > > Thanks for your support!
> > > We have updated a new version of our paper, where we add more discussion about our motivations, and differences with related work throughout the paper (all marked with blue). We hope this version further resolves your concerns.

---

> > > > ### Comment · Reviewer_Ydk8 · 2023-12-03
> > > >
> > > > Thank you. I have read the new version.

---

### Official Review · Reviewer_5Dh6 · 2023-11-01

**Soundness:** 2 fair
**Presentation:** 2 fair
**Contribution:** 2 fair
**Rating:** 5
**Confidence:** 1

**Summary:**

The authors delve into an investigation of the impact of token repetition on in-context learning (ICL) performance, revealing both beneficial and detrimental effects through empirical analysis. On the positive side, repetition aids in narrowing down the output space, fostering consistency in subsequent predictions. However, the downside is evident when non-informative or incorrect messages are repeated in the demonstrations, resulting in diminished prediction accuracy.

**Strengths:**

The authors conducted extensive experiments to explore the nuances of token repetition and its effects on ICL.

**Weaknesses:**

1. **Lack of Clear Motivation**: The rationale behind investigating token repetition patterns is not adequately clear, leaving the reader uncertain about the study's purpose. Additionally, the abstract does not effectively capture the paper's core idea, necessitating further clarification to provide a concise summary of the work.
2. **Need for Enhanced Clarity**: The manuscript's presentation is complex, making it challenging for readers to navigate through the content. A more detailed explanation of certain terms and concepts, such as *surface pattern*, *self-reinforcement effect*, and *token reinforcement*, would significantly enhance comprehension, particularly if these have been previously established in prior literature.
3. **Specificity of Experiments**: The experiments are overly specific and do not sufficiently connect to real-world applications. Demonstrating how the findings on repetition patterns can be applied in practical scenarios would strengthen the paper's relevance and impact.

**Questions:**

While I do not have any specific questions at this moment, I would appreciate it if the authors could address the concerns raised in the Weakness section, particularly regarding the need for clearer motivation and enhanced clarification of certain terms and concepts.

---

> ### Author Response · Authors · 2023-11-21
> **Author Response to Reviewer 5Dh6**
>
> Thanks for reviewing our paper.
>
> We provide three additional experiments, aiming to resolve the reasons of token reinforcement: (1) The similar token reinforcement lies in the pretraining dataset and thus it is learned by LLMs; (2) repeating features responsible for token reinforcement are effective in the pre-training stage, and thus would be utilized by LLMs; (3) Attending preceding token is responsible for reinforcement.
>
> In addition, we clarify our contributions and implications and refer you to our general response and newly revised paper(with revised part highlighted with blue).
>
> We add the above mentioned experiments to the paper and promise to update the main content accordingly. We hope these results resolve your concern and will be really appreciated if you re-consider the score of our paper.

---

### Official Review · Reviewer_iqcg · 2023-11-02

**Soundness:** 3 good
**Presentation:** 3 good
**Contribution:** 2 fair
**Rating:** 5
**Confidence:** 4

**Summary:**

This paper studies the emergent in-context learning (ICL) ability of LLM. The authors try to probe the ICL performance through surface repetitions, and establish a theory called token co-occurrence reinforcement, which explains the reasons for possible failures.

**Strengths:**

- This work provides meaningful explanations on the possible failures of ICL.
- The experiments seem comprehensive and convincing.

**Weaknesses:**

- This work tries to understand the inherent ICL behavior of LLMs, yet is in lack of theoretical analysis. For example, how is such token co-occurrence reinforcement established? This may involve the detailed interactions between prompts and self-attention mechanism, etc., which I would like the authors to delve into.
- As an experiment-oriented work, the authors should examine their assumptions on more LLMs; otherwise, it's hard to reach a common conclusion.
- The findings of this work are not completely new. As far as I'm concerned, the findings are based on the distributional bias in the demonstrations. The impact of spurious correlations are widely discussed in out-of-distribution generalization literature. In this regard, more insights are welcome. Also, the authors could discuss on how to address such (inevitable) distributional bias in the demonstrations.

**Questions:**

As mentioned above, how could me mitigate the biased effect in ICL demonstrations?

---

> ### Author Response · Authors · 2023-11-21
> **Author Response to Reviewer iqcg**
>
> Thanks for your time and insightful reviews.
>
> **Q: “how is such token co-occurrence reinforcement established?”**
>
> A: We provide three additional experiments, demonstrating that (1) The similar token reinforcement lies in the pretraining dataset and thus it is learned by LLMs; (2) repeating features responsible for token reinforcement are effective in the pre-training stage, and thus would be utilized by LLMs; (3) Attending preceding token is responsible for reinforcement.
>
> Based on these experiments, we explain the inevitable reason of token reinforcement is the models’ effort to maximize training likelihood.
>
> For more details, we refer you to the general response and Appendix C in our revised version of paper.
>
> **Q: “examine assumptions on more LLMs”**
>
> A: Thanks for your advice! We further consolidate our findings on LLaMA-2(7,13,70b), GPT-J and Mistral. We now examine 16 LLMs in total. All results (Appendix D.2, marked as blue) accord with our findings in main content.
>
> **Q: “distributional bias in the demonstrations”:**
>
> A: This is an insightful perspective to understand our work.
>
> In our understanding, our findings provide an interesting perspective of distributional bias.
>
> If we consider the distributional bias between the pre-training stage and testing, in-context learning itself should be regarded as a result of distributional bias. In this regard, we show that this distributional bias is helpful through token reinforcement, where the distributional bias studied in OOD is generally harmful. The beneficial part of the bias should be enhanced, and the detrimental part should be mitigated.
>
> We would like to discuss more related work of distributional bias in our next version.
>
> **Q: “How to mitigate such bias?”**
>
> A: Currently, we do not have an experiment showing how to resolve this. Our intuition is that such reinforcement indicates concentration in the representation space at certain direction. If we can detect the direction, we can selectively enhance or mitigate the beneficial and detrimental parts of the bias.
>
> We add the above mentioned experiments to the paper and promise to update the main content accordingly. We hope these results resolve your concern and will be really appreciated if you re-consider the score of our paper.

---

> > ### Comment · Reviewer_iqcg · 2023-11-22
> >
> > I appreciate the authors' tremendous efforts made in rebuttal, which has addressed some of my concerns, and I will re-consider the score after discussing with other reviewers.
> >
> > If I understand correctly, this paper proposed a new perspective that tries to consider ICL as the combination of a series of token reinforcement effects that have been potentially established during LLM pre-training, and can be triggered with few in-context demonstrations. This perspective is sufficient since in pre-training, LLM already sees every word requisite for the generative inference, and hence there can always be some "token reinforcement" effect which connects some pre-trained word (or the in-context word) to the generated results. However, as far as I'm concerned, it cannot necessarily prove that token reinforcement is the very cause that enables LLM to understand the intended task given only several demonstrations.
> >
> > Regarding the distributional bias, I believe that the biases discussed in OOD literature and in this paper are essentially the same. A bias is considered "good" when it is in line with the desired pattern in inference, and hence there is no bias that is purely "good" or "bad".
> >
> > In summary, I believe that the proposed token reinforcement effect can indeed explain some common behaviors of ICL, but I feel that the actual working mechanism of ICL is beyond token reinforcement.  I would also welcome the authors' and the other reviewers' further input in case I have missed something.

---

> > > ### Author Response · Authors · 2023-11-22
> > >
> > > Thanks for the affirmation of our efforts.
> > >
> > > One clarification we want to make is that we are not arguing that token reinforcement is the only reason for in-context learning. Our new perspective takes one step further in learning the cause of ICL.
> > > As you mentioned, the experimental result and findings in our paper show that token reinforcement is one of the important feature in ICL, responsible for various behaviors of ICL.
> > > We believe this is a necessary step towards unraveling the puzzle of in-context learning and motivating further research into its underlying principles. We hope this clarification will resolve your concern.
> > >
> > > Regarding distributional bias, we think we are on the same page now. Whenever the bias leans toward unwanted behaviors, we want to mitigate such biases. Following our discussion, we will review distributional bias in OOD literature to the related work section in our revision.

---

### Official Review · Reviewer_d42T · 2023-11-02

**Soundness:** 3 good
**Presentation:** 3 good
**Contribution:** 3 good
**Rating:** 6
**Confidence:** 4

**Summary:**

This paper quantitatively investigates in-context learning in terms of surface patterns.
It shows that there is an inherent correlation between surface patterns, self-reinforcement, iterative generation and their important role in text generation.
In particular, it shows the role of surface surface patterns in text generation and the existence of token co-occurrence reinforcement that strengthens the relationship between two tokens based on their contextual co-occurrences.
The experiments on MMLU and GSM8K show that the reinforcement helps to constrain the output space and format the output according to a  demonstration such as ‘Let’s think step by step’ .

**Strengths:**

This paper provide a novel framework to understanding in-context learning via  the notion of token co-occurrence reinforcement.
Through various experiments, the authors have shown how token reinforcement causes spurious correlations in in-context learning.

**Weaknesses:**

Althogh  there is a novelty in showing experimentally that token reinforcement can cause some problems in in-context learning,
the paper lacks the important perspective of analyzing why token reinforcement exists and causes problems.
For example, the following paper, which is only briefly mentioned in this paper, analyzes the impact of repetition structures in a corpus on in-context learning from an information-theoretic perspective.
A Theory of Emergent In-Context Learning as Implicit Structure Induction
Michael Hahn, Navin Goyal
They showed that the performance of in-context learning is represented by a complexity that repetition structures can be represented by a small PCFG tree, and  experimentally investigated theoretical finindings.

**Questions:**

Is it possible to provide hypothesis about the reason for the token reinforcement phenomenon and check it, even experimentally?
The authors may argue that analyzing reasons is the next step, i.e., outside the scope of this paper; however, it is fandamental in machine learning research.

---

> ### Author Response · Authors · 2023-11-21
> **Author Response to Reviewer d42T**
>
> Thanks for your time and insightful reviews.
>
> **Q: “the reason for the token reinforcement effect”:**
>
> A: We provide three additional experiments, demonstrating that (1) The similar token reinforcement lies in the pretraining dataset and thus it is learned by LLMs; (2) repeating features responsible for token reinforcement are effective in the pre-training stage, and thus would be utilized by LLMs; (3) Attending preceding token is responsible for reinforcement.
>
> Based on these experiments, we explain the inevitable reason of token reinforcement is the models’ effort of maximizing training likelihood.
>
> For more details, we refer you to the general response and Appendix C in our revised version of paper. (At the first glance, we really gonna say it’s the next step of this paper. However, we reckon that this perspective is an important complement for our paper and we are also very interested in the reasons.)
>
> **Q: “related work”**
>
> A: Thanks for pointing out the related work! We have updated our related work section.
>
> We add the above mentioned experiments to the paper and promise to update the main content accordingly. We hope these results resolve your concern and will be really appreciated if you re-consider the score of our paper.

---

> > ### Comment · Reviewer_d42T · 2023-11-22
> >
> > The author's reply and additional experiments seem interesting to me.
> > Thus, I raise my score.

---

> > > ### Author Response · Authors · 2023-11-22
> > >
> > > Thanks for your support! We are glad that you find our new experiments interesting.

---

### Author Response · Authors · 2023-11-21
**General Response to All Reviewers**

To all reviewers,

We thank all the reviewers for taking the time to review our paper and their valuable comments. We are grateful that reviewers appreciate the effort of our work in analyzing the In-Context Learning of LLMs from a new perspective with comments such as Reviewer d42T "novel framework", Reviewer iqcg "meaningful explanations, comprehensive and convincing", Reviewer Ydk8 "systematic study, interesting results”.

We notice that a common concern raised by reviewers is the cause and inner-working responsible for the token reinforcement.

We have delved into the reasons underlying token reinforcement and conducted three important experiments to enhance our paper.

**As for the reasons, we believe that token reinforcement could be a result of the model's effort to maximize the likelihood of the training data.** Token reinforcement is hidden in the nature of pre-training corpus.  In light of this, we conduct experiments to validate our hypothesis.

Given a commonly used large pre-training corpus (i.e., wikipedia-english(https://huggingface.co/datasets/olm/olm-wikipedia-20221220), 5.7B tokens), we compute the following statistical probabilities:

Exp_w P(w occurs | w is never observed in the context)

Exp_w P(w occurs | w has been observed before in the context for 1 time)

Exp_w P(w occurs | w has been observed before in the context for 2 times)

…

where w is any word, and context is the autoregressive context window of pretraining data, usually 1024 tokens.

Conceptually, these probabilities reveal whether a particular word $w$ is likely to recur if there have already been $n$ instances of $w$ observed in the context, resonating with our analysis of token reinforcement.

The statistical probabilities are shown below (n=[0,9]):

| Num Occurrence | Probabilities | Value |
| --- | --- | --- |
| 0 | █░░░░░░░░░░░░░░░░░░░ | 1.32% |
| 1 | ██████░░░░░░░░░░░░░░░ | 19.53% |
| 2 | █████████░░░░░░░░░░░░ | 35.63% |
| 3 | █████████████░░░░░░░░ | 47.95% |
| 4 | ███████████████░░░░░░ | 53.26% |
| 5 | █████████████████░░░░ | 57.73% |
| 6 | ██████████████████░░░ | 59.15% |
| 7 | ███████████████████░░ | 60.48% |
| 8 | ████████████████████░ | 60.78% |
| 9 | ████████████████████▒ | 60.79% |

The probabilities of a specific word occurring with an increasing number of instances perfectly align with the token reinforcement on LLMs.

The LLM manages to learn this inherent feature from the training data and generalize it to longer phrases and distant connections. This also elucidates the scaling with reinforcement, where larger models more effectively capture this feature from the training corpus.

To further understand the existence and inner workings of token reinforcement, we provide more experiments from two perspectives. The first one shows from learning process that **leveraging repeating features are effective in the maximizing likelihood**. We pretrain three conditional LMs from scratch and compare the converged loss of three strategies: (1) full context (2) random mask (3) mask repetitive features. Our results show that (3) mask repetitive features achieves the worst converged loss, which validates our hypothesis.
The second experiment analyzes the trained LLMs and shows that the **attention mechanism of attending preceding/adjacent token is responsible for the inner-workings of token reinforcement**. After masking out the adjacent token, the strength of reinforcement is reduced by 10 times.

For more details and experimental settings, we refer all reviewers to **Appendix C**. We highlight the revised parts of the paper with blue.

**Finally, our main contributions are:**

1. We propose a novel perspective to understand icl with repetitive text generations, where the key effect causing repetitions appears to be an important feature in ICL.
2. We empirically establish the existence of token reinforcement across various LLMs, alongside with the inevitable reason behind learning process and inner workings responsible for it.
3. We show that token reinforcement plays crucial role in ICL for both beneficial and detrimental effects. It constrains output space and desired patterns, but it is also responsible for spurious connections and possible failure of ICL. Our findings provide a novel research basis for analyze the mechanism of in-context learning.

*Based on our results, instead of the traditional understanding of input-label mappings, we view ICL as a combination of token reinforcements, even though some of them only appear once and do not get reinforced much.*

Due to the time limit in the discussion period, we are committed to incorporating these new results about the reasons and inner workings of token reinforcement into the main content, as well as the highlighting our contributions and implications in the next version.

We hope we have addressed all your concerns and would greatly appreciate it if you could re-consider your score about our paper based on our responses.

---

> ### Comment · Reviewer_Ydk8 · 2023-11-21
>
> Thanks to the authors for taking the time for these additional results. I agree that they can address to  some extent the concerns about lack of clarity about contribution raised in my review. I’ll reconsider my scores based on this.

---

> ### Author Response · Authors · 2023-11-22
> **Update the main content of our paper**
>
> We have updated the main content of our paper, including revisions in the introduction, additional experiments about reasons and explaining the implications, and discussions about how our studies can guide the selection of demonstrations. All revisions are marked blue. We hope this version is a clearer presentation of our work.
>
> Thanks for your consideration.

---

### Meta-Review · Area_Chair_kXrk · 2023-12-05

**Metareview:**

This paper investigates an interesting aspect of the in-context learning ability of LLMs. They show that LLM behaviors are sensitive to the repetitive demonstrations in the context. Built upon this observation, they take a close analysis on the beneficial and detrimental effects of reptition through empirical analysis. On the positive sides, repetition helps constrain the output space and enhance consistency; on the downside, repeating wrong information can be very harmful and leads to incorrect answers. Extensive experiments are carried out to justify these claims.

The reviewers generally find the problem interesting and delivers new insights to the community. During rebuttal, some concerns rise on the lack of theoretical explanations of the discovered phenomena, and the authors provide some insights with new experiments. Another reviewer point a limitation of novelty since repetition is only a special case of distribution shift, which does make sense since repetition brings change to the distribution by upweighting some examples.

Summarizing these opinions, I find the discovered phenomena reveal certain risks of LLMs and the authors have put great efforts to empirically understood their behaviors and influence on LLM outputs. Given the timely interest in LLMs in the field, I would recommend accepting this paper to facilitate better understanding. I highly encourage the authors to incorporate the new results in the final version.

**Justification For Why Not Higher Score:**

This paper is mainly empirical and lacks a deeper understanding of the discovered behaviors. The studied subject, repetition, is also a special case of distribution shift, that limits its generality.

**Justification For Why Not Lower Score:**

This paper provides a detailed investigation on the benefits and downsides of the repetition of LLMs, which could inspire better understanding as well as new measures to solve these problems.

---

### Decision · Program_Chairs · 2024-01-16

Accept (poster)